Shining a light on duckweed: exploring the effects of artificial light at night (ALAN) on growth and pigmentation

Nakagawa-Lagisz Totoro totoro22ro@gmail.com 1 2
Lagisz Malgorzata losialagisz@gmail.com 3 4
1 Sydney Boys High School , Sydney , NSW , Australia
2 Onna Junior High School , Onna , Okinawa , Japan
3 Evolution & Ecology Research Centre and School of Biological, Earth & Environmental Sciences, University of New South Wales Sydney , Sydney , New South Wales , Australia
4 Theoretical Sciences Visiting Program, Okinawa Institute of Science and Technology Graduate University , Onna , Okinawa , Japan
Khan Mather
Electronic publication date: 2023 Nov 1
Publication date: 2023
Volume: 11
Electronic Location ID: e16371
Received 2023 Jun 12; Accepted 2023 Oct 8
Copyright: ©2023 Nakagawa-Lagisz and Lagisz
Copyright year: 2023
Copyright holder: Nakagawa-Lagisz and Lagisz
License: This is an open access article distributed under the terms of the Creative Commons Attribution License, which permits unrestricted use, distribution, reproduction and adaptation in any medium and for any purpose provided that it is properly attributed. For attribution, the original author(s), title, publication source (PeerJ) and either DOI or URL of the article must be cited.
License URL: https://creativecommons.org/licenses/by/4.0/

Keywords: Abiotic stress, Light stress, Anthropocene, Urbanisation, Anthropogenic effects, Plant physiology, Urban adaptation

Funding: The authors received no funding for this work.

==============================
Background

Artificial light at night (ALAN) is a novel environmental stressor of global concern. Various sources of artificial light are now common in urbanized areas and have diverse negative effects on many species of animals and plants. However, ALAN has also been shown to have no effect or a positive effect on some organisms. This study investigates the impact of ALAN on the growth and leaf pigmentation of a common floating freshwater plant species.

Methods

We exposed wild-derived dotted duckweed (Landoltia punctata) to either darkness during the night (Control group) or to artificial light at night (ALAN group) for 49 days. We set up two large boxes of eighty samples each with 2–3 leaves of duckweed in each sample at the start of the experiment. The ALAN box had an opaque lid with a small lamp that was turned on at night. The Control box was also covered at night with an opaque lid but without a lamp. During the day, plants in both boxes were exposed to natural light. We counted the number of leaves in each sample weekly. We took photos of the samples on day 28 to measure the total leaf surface area per sample. On day 49, we took photos of the underside of the leaves for analyses of the relative levels of dark pigmentation across all samples.

Results

We found that ALAN-exposed plant samples had, on average, more leaves than control plants after a few weeks of exposure. They also had a more variable number of leaves per sample. The total leaf area per sample on day 28 was larger in the ALAN samples. The underside of the leaves on day 49 was, on average, darker in the ALAN plants than in the control plants.

Conclusion

There is a significant growth-enhancing effect from exposure to artificial light at night on Landolita punctata. However, higher variability induced by ALAN exposure indicates that ALAN is also a stressful condition for these plants. This is in line with our finding of the presence of larger amounts of dark pigments in the leaves of ALAN-exposed plants. Dark pigmentation in duckweed species could be a defence mechanism protecting tissues from stress-induced oxidative damage. Overall, both positive and negative effects of ALAN can be observed simultaneously in different traits of the same organism. Increased individual variation can facilitate population-level adaptation to stressful conditions. As such, this work contributes to our knowledge of the effects of light pollution in urban environments on common plants.

Introduction

In urbanized areas, human interference has altered natural ecosystems. One of the human-induced changes is the common presence of artificial lights that are used at night, especially in urban areas. The sources of artificial light include streetlights, lights in houses, car lights, as well as garden and park lamps. Artificial light at night (ALAN) can impact all types of living beings, with effects cascading to populations and ecosystems (Davies, Bennie & Gaston, 2012; Gaston et al., 2013; Hölker et al., 2015). Plants are also vulnerable to the effects of artificial lights because plants in urban areas are constantly near such lights, which has been shown to have negative effects on plant pollination and physiology (Knop et al., 2017; Murphy et al., 2022).

Artificial light at night can be especially harmful to plant growth. Growth can be inhibited because during the dark phases of the daily light cycle plants use accumulated solar energy for photosynthesis (Sukhova, Vodeneev & Sukhov, 2021). Thus, while short periods of ALAN, like keeping a garden light on in the evening, may not have any biologically significant effects on the plants, more prolonged or intense exposure may cause stress, disrupting the biological clock, signalling pathways and growth (Eriksson & Millar, 2003; Farré, 2012; Roeber et al., 2021).

Duckweed has been long popular in research on plant physiology because of their small size, ease of culturing and fast growth (Hillman, 1961; Hillman, 1976; Strzałek & Kufel, 2021). Duckweeds (Lemnoideae) comprise 36 species and are considered to be the smallest plants among Angiosperms (Laird & Barks, 2018; Baek, Saeed & Choi, 2021). They usually self-replicate vegetatively by sprouting new fronds (leaf-like assimilatory organs) from the existing fronds, so that each frond is a clonal plant (Ziegler et al., 2015). Notably, the term “frond” can be also used interchangeably with “leaf”. While many studies investigated the effects of photoperiod on duckweed growth, they were conducted in laboratory conditions with plants exposed exclusively to artificial lights. Results of such studies were often mixed, with some reporting no effects (e.g., Gallego, Chien & Angeles Jr, 2022) and others reporting increasing growth rates with longer light exposures (e.g., Yin et al., 2015).

Plants, including duckweed species have been shown to respond to stressful abiotic conditions, including high light intensity, by producing a range of antioxidants to prevent tissue damage (Baek, Saeed & Choi, 2021). Under stressful high-light conditions, chloroplasts generate reactive oxygen species (ROS) which in turn can trigger anthocyanin accumulation (Ma et al., 2021). Anthocyanins can be observed as red-orange to blue-violet pigmentation in many plants (Wallace & Giusti, 2015), manifesting as brownish / purplish leaves or red fruit.

Anthocyanins have been shown to reduce the ROS levels in plant tissues, allowing plants to maintain photosynthetic capacity (Kovinich et al., 2015; Xu, Mahmood & Rothstein, 2017). However, under extreme stress conditions, the protection by anthocyanins and other antioxidants may be insufficient and ROS can still cause oxidative damage to the plant tissues, resulting in reduced growth and reproduction (Naing & Kim, 2021).

Duckweed fronds exposed to abiotic environmental stressors, namely heavy metals and organic pollutants, exhibit elevated levels of anthocyanins (Jayasri & Suthindhiran, 2017; Sharma & Kaur, 2020), implying these dark pigments play a role in stress defence also in this group of species. Night irradiation with LED lights was linked to anthocyanin accumulation in the skin of grape berries (Kondo et al., 2014) and in broccoli sprouts (Kohda et al., 2022), suggesting that the ALAN is a stress factor capable of inducing increased anthocyanin production.

In this project, we study the effects of artificial light at night (ALAN) on duckweed plants exposed to natural light conditions during the day. We aimed to answer the following two research questions: (1) How is the growth of duckweed plants affected by exposure to ALAN? We hypothesize that exposure to ALAN will reduce plant growth. (2) How is the leaf colour (pigmentation) affected by exposure to ALAN? We hypothesize that exposure to ALAN will induce increased production of dark pigments, resulting in a darker leaf colour.

Materials & Methods

Plant collection and species description

We collected a large sample of floating duckweeds on 25th February 2022 from Busby’s Pond located in Centennial Park, Sydney, NSW, Australia (coordinates: −33.897873 151.228606). We placed the plants into plastic buckets with the pond water (pH = 7.4, Ammonia = 1.0 ppm, Nitrate = less than 0.25 ppm, Nitrite = less than 1.0 ppm; API Freshwater Master Test Kit) and allowed collected plants to acclimate to indoor conditions for 4 days.

After examination under a bifocal microscope, we determined the duckweed sample species to be Landoltia punctata (Les & Crawford, 1999). This species is native and common in New South Wales (PlantNET, 2023; NSW FloraOnline; https://plantnet.rbgsyd.nsw.gov.au/; accessed March 2022). The typical frond (a clonal plant) is made from one to three leaves: a mother leaf, a budding out new leaf, and sometimes also another new buddying leaf. The leaves are very variable in terms of their size, shape, and pigmentation (Les & Crawford, 1999; Baek, Saeed & Choi, 2021). The upper side of leaves is typically dark green, but the lower side can accumulate red pigment and appear dark brown or purplish (Kittiwongwattana, 2019).

Experimental setup

Figure 1A summarizes the experimental design. We conducted the experiment using 161 30 mL clear plastic cups. we filled all cups with 20 mL of pond water each. We labelled these cups with a number 1–161 and drew a water level line at 20 mL for future refills. We placed one randomly selected duckweed frond (2–3 connected leaves) into each cup. Then, we used an R script to randomly assign a cup into either the Control (N = 81) or ALAN (N = 80) group. We placed the cups into one of the two large holding boxes (Control and ALAN) described below.

Figure 1 Experimental setup.

Wild-derived duckweed plants were separated into individual fronds/plants (2–3 connected leaves) and assigned at random to ALAN (artificial light at night) or Control (natural photoperiod) conditions for 49 days. Weekly leaf counts were conducted to track duckweed growth and reproduction. On Day 28, images were taken to measure the total leaf area, and on Day 49 images of the underside of the leaves were used to quantify the relative amounts of dark pigmentation.

The two holding boxes were 50 L in volume, made of black opaque plastic and had cardboard lids fitted on top to block any outside light at night, but allowing ventilation via a small gap under the overhanging edges of the lids (Fig. 1B). One box was used as a Control condition, and the other as an ALAN condition. The lid that covered the ALAN box had a small hole in it which was mounted with a lamp (LED lamp, Correlated Color Temperature (CCT) 3000K) for shining artificial light at night on ALAN-group plants. We placed the two boxes next to each over near a west-facing window (Fig. 1B), approx. 10 cm from the fully transparent large window glass panel and 6 m from an air conditioner located at the top of an internal wall. We swapped positions of the two boxes every few days.

At the beginning of each day, we took the lids off the boxes, and the lamp was switched off. At dusk, we put the lids back on, and turned the lamp on. Every few days, we refilled water lost to evaporation in the cups, as needed, and randomly rearranged the cups within each box to negate potential small differences in the light exposure or temperature within each box. Every few weeks, we measured the temperature, humidity, and light intensity with a digital thermometer, humidity meter, and a Photon app (https://growlightmeter.com/; v.0.4.2 for iPhone SE; free version downloaded March 2022), within each box at a few different locations.

The experiment was conducted between 27 February and 17 April 2022, which corresponds to the early autumn season at the experimental location. The exact geographic location of the experiment was: latitude of −33.92296868040449 and longitude of 151.2373484125205 (coordinate values derived from Google Maps).

During the day, the amount of light illumination in both boxes was, on average, 493 lux (Standard Deviation, SD = 144 lux). During the night, the illumination in the ALAN box was, on average, 40.5 lux (SD = 8 lux). During the night, the illumination in the Control box was, on average, 0.2 lux (SD = 0 lux). The difference in illumination between the Control and ALAN boxes at night was statistically significant (Wilcoxon test: W = 16, p = 0.021).

To answer our research questions, we estimated duckweed growth as weekly leaf counts per sample and as total leaf surface area per cup on Day 28. On Day 49, we measured leaf pigmentation levels as a percentage of dark-coloured leaf area on the underside of the leaves, where most red pigment accumulates in this duckweed species (Fig. 2). We provide detailed descriptions of these measurements below.

Leaf counts

During the experiment, we weekly counted the number of live leaves per cup in all the cups to collect data on duckweed growth and reproduction (via leaf buddying). Before measurements, one researcher removed the labels from the boxes so that the other researcher counting leaves would not know about the treatment group identity. Such “blinded” data collection aims to reduce unconscious bias during measurements.

One researcher wrote down the number of leaves and the corresponding cup number, as well as refilled the pond water in the cup as needed. Once we completed processing one box, the other researcher shuffled the cups around and placed them back in their original box. Then, we repeated the process with the other box. When the counting was finished, we placed the labels of each box back on, and entered the count data into a spreadsheet. We then cross-checked the entered data for errors and corrected any mismatches by re-checking cups as needed.

On experiment Day 42 and Day 49, along with the number of live leaves, we also recorded the number of dead leaves. One (blinded to the treatment group) researcher counted the number of discoloured (i.e., turned white / yellow, or completely falling apart) leaves that were floating on the water’s surface. Such dead leaves were rare before Day 42, and because of this, they were initially not counted. We then calculated total leaf counts for Day 42 and Day 49, by adding counts of live and dead leaves together, to represent the rates of growth accounting for duckweed aging and mortality.

Figure 2 Example duckweed photos for sample number 32.

(A) Upper side of duckweed leaves from the sample on Day 28: image used to quantify the total leaf area per sample. (B) The underside of the leaves from the same sample on Day 49: the image used to quantify the relative areas with high contents of dark pigments. Both images were cropped to remove the excess background.

Leaf surface area

On Day 28 of the experiment, we measured the total surface area of leaves within each cup. To do so, we built a cardboard stand that was set up to hold a phone firmly so that it would not shake when taking photos. We used a small ceramic base plate as a white background for taking plant images. The plate had four black dots in a square shape, each three cm apart. This was to allow the iPhone app LeafByte (Getman-Pickering et al., 2020) to accurately calculate the total leaf surface area by scaling it to the dot locations. To correctly identify each plant sample in the photo taken, we wrote the ID number of the sample with an erasable pen just outside the square formed by the four dots, within the camera vision range. We then gently removed all the plants, one by one, out of their cups (with the use of a thin brush), with a small amount of water to prevent desiccation-related damage(which occurs rapidly in small aquatic plants). We placed all leaves within the four dots area to take a photo, with the upper side of the leaves facing up. We took a photo of each sample. After this, we quickly returned the plants to their corresponding cup using a brush. We repeated this process for every experimental cup until we had a photo for every sample. We then used the LeafByte app to calculate the total leaf area of each sample and exported all the data from the app to an Excel spreadsheet.

Leaf pigmentation

We estimated pigmentation levels of the undersides of the leaves on Day 49 of the experiment. We had the same setup as for the measurements of the total area of the leaves on Day 28 but used a flat background plate, avoided transferring excess water for better image quality, and flipped the leaves upside down so that the bottom side of the leaves (which accumulates most anthocyanins (Kittiwongwattana, 2019)) was facing up. We analysed the images using an R package pliman v. 1.1.0 (Olivoto, 2022). This package can extract the colouration of the leaves by removing the background and classifying areas with different pigmentation levels into two classes “dark” (more pigments) and “light” (less pigmented). The algorithm is based on a general linear model (binomial family) fitted to the RGB values of each image, calibrated on a set of predefined color palette samples. This approach took into account that, within each sample, some leaves (or leaves parts) could be less pigmented, and others could be more pigmented. In our workflow, for each duckweed sample, we estimated the proportion of total leaf area classified as “dark”.

Statistical analyses

We conducted all statistical analyses in the R v.4.2.3 statistical environment (R Core Team, 2021). Analysis code and processed data are available on GitHub (https://github.com/lordofthepigs123/ALAN_duckweed and archived on Zenodo with doi:10.5281/zenodo.8017670). We compared average light exposure for each group of samples, time trends and differences in weekly leaf counts during the experiment, total leaf area on Day 28 and pigmentation on the underside of leaves on Day 49. Sample sizes were initially 80 for ALAN and 81 for Control group, with the first 80 from each group used for weekly counts. Two samples of plants were lost during taking photo measurements due to spilling them on the floor. Missing values were omitted from statistical analyses; thus, sample sizes are 80 and 80, per replicate, for most statistical tests.

Leaf counts across 7 weeks

The main model for duckweed growth was based on the weekly counts of leaves over 7 weeks. We used an R package report v.0.5.7.4 (Makowski et al., 2023) to summarize statistical models: We standardized the experimental day variable into z-score (resulting in a distribution with a mean of 0 and standard deviation of 1). Using package lme4 v.1.1-31 (Bates et al., 2015), we then fitted a Poisson mixed model (estimated using ML and Nelder–Mead optimizer) to predict changes in leaf counts within each experimental group (formula: growth ∼1 + group + zday) across duration of the experiment. The model included sample identity (cup/sample number) as a random effect(formula: ∼1 — sample). We obtained standardized parameters, including 95% Confidence Intervals (CIs) and p-values using a Wald z-distribution approximation. Additionally, we used the Wilcoxon rank sum test and variance test (F-test) for each weekly leaf count separately to compare medians and variabilities in leaf counts on a given week (R package stats v.4.2.3; environment; R Core Team, 2021). We applied the same two statistical tests to the number of dead leaves and total numbers of leaves on Day 42 and Day 49.

Total leaf surface area and pigmentation

We compared leaf total surface area on Day 28 between two experimental groups using the Wilcoxon rank sum test and compared the amount of variation within each experimental group using the variance test (F-test). We applied the same two statistical tests to the numbers representing the proportions of areas with dark pigmentations underneath the leaves measured on Day 49. We also conducted exploratory analyses of the correlation between leaf count and total leaf area on Day 28, and of leaf count and proportion of dark pigmented area on Day 49 (package stats v.4.2.3, environment; R Core Team, 2021).

Results

Leaf counts across 7 weeks

Using a Poisson mixed model predicting leaf counts within experimental cups across 7 weeks, we found a tendency for a positive effect of ALAN treatment on leaf counts (estimated difference between ALAN and Control = 0.079, 95% CI [−0.01, 0.18], p = 0.079; Standard Error, SE = 0.04, 95% CI [−0.005, 0.09]), as shown in Fig. 3. The number of leaves increased over time (slope = 0.38, 95% CI [0.36, 0.40], p < .001; SE = 0.38, 95% CI [0.36, 0.40]). The model’s total explanatory power was substantial (conditional R 2 = 0.62, marginal R 2 = 0.41; sensu Nakagawa & Schielzeth, 2013).

Duckweed plants had the same number of live leaves on Day 0 (start of the experiment) in ALAN and Control groups (Wilcoxon rank sum test: W = 3170, p = 0.906). From Day 7, there was a tendency for more live leaves in the ALAN group than in the Control group, but it was not consistently statistically significant across all weeks (Table 1).

The mortality, estimated as the number of dead leaves, was greater in the ALAN group on Day 42 (W = 4920. p = 0.000) and Day 49 (W = 4830, p = 0.000) (Figs. 4A and 4B). Total counts of leaves (live and dead) revealed higher overall growth and reproduction in the ALAN than in the Control group on Day 42 (W = 3808, p = 0.037; Fig. 4C), and a tendency in the same direction on Day 49 (W = 3745, p = 0.061; Fig. 4D).

The leaf numbers in the ALAN group were increasingly more variable than in the Control group as the experiment progressed (Table 1), especially on Day 49 (variance test: F = 2.007, p = 0.002, df = 79, 79). This pattern of greater variability in the ALAN group was also clear for the total counts of live and dead leaves on Day 42 (F = 1.886, p = 0.005, df = 79, 79) and Day 49 (F = 2.193, p = 0.001, df = 79, 79).

Total leaf surface area and pigmentation

Total leaf area for each sample measured on Day 28 showed a clear difference (W = 4020, p = 0.005; Table 1), indicating that plants in the ALAN group grew and produced new fronds faster than the Control group (Fig. 4E). As with the leaf counts, the leaf area of plants in the ALAN group was more variable than in the Control plants on Day 28 (F = 1.814, p = 0.009, df = 79, 79). Proportion of the underside leaf area with dark pigment content on Day 49 was higher in the ALAN group plants than in the Control plants (Fig. 4F; W = 4738, p = 0.000; Table 1), with higher variance observed in the ALAN group (F = 0.628, p = 0.041, df = 78, 79).

Figure 3 Weekly counts of live duckweed leaves of 80 ALAN (artificial light at night) samples and 80 Control (natural photoperiod) samples.

Darker points with whiskers represent mean values for each group and their confidence intervals. Figure was created using the R package ggplot2 v.3.4.2 (Wickham, 2016) and ggbeeswarm v.0.7.2 (Clarke, Sherrill-Mix & Dawson, 2023).

Table 1 Results of statistical tests comparing medians and variances for measurements taken at different time points (Days) during the experiment.

Nonparametric Wilcoxon rank sums test was used for comparing medians. Variance test was used for comparing variability. Dample sizes were 80 for ALAN (artificial light at night) and 80 for Control (natural photoperiod) duckweed samples.

Day	Measurement	Comparing medians	Comparing variances	df	
		W	p	F	p	num.	denom.	
0	Live leaf count	3170	0.906	1.03	0.878	79	79	
7	Live leaf count	3810	0.031	0.85	0.473	79	79	
14	Live leaf count	3350	0.602	1.46	0.095	79	79	
21	Live leaf count	3730	0.070	1.46	0.096	79	79	
28	Live leaf count	3750	0.059	1.90	0.005	79	79	
35	Live leaf count	3460	0.369	1.95	0.003	79	79	
42	Live leaf count	3460	0.375	1.89	0.005	79	79	
49	Live leaf count	3360	0.596	2.01	0.002	79	79	
42	Dead leaf count	4920	0.000	1.42	0.121	79	79	
49	Dead leaf count	4830	0.000	2.16	0.001	79	79	
42	Total leaf count	3810	0.037	1.89	0.005	79	79	
49	Total leaf count	3740	0.062	2.19	0.001	79	79	
28	Live total leaf area	4020	0.005	1.81	0.009	79	79	
49	Proportion dark area	4740	0.000	0.63	0.041	78	79	

Figure 4 Comparisons of 80 ALAN (artificial light at night) and 80 Control (natural photoperiod) duckweed samples at various time points during the experiment.

(A) Number of dead duckweed leaves for ALAN (artificial light at night) and Control (natural photoperiod) samples for experimental Day 42. (B) The number of dead duckweed leaves for ALAN (artificial light at night) and Control (natural photoperiod) samples for experimental Day 49. (C) Total number of live and dead duckweed leaves for ALAN (artificial light at night) and Control (natural photoperiod) samples for experimental Day 42. (D) Total number of live and dead duckweed leaves for ALAN (artificial light at night) and Control (natural photoperiod) samples for experimental Day 49. (E) Total leaf area of live duckweed leaves for ALAN (artificial light at night) and Control (natural photoperiod) samples on experimental Day 28. (F) Proportion of the dark pigmented underside of live duckweed leaves for ALAN (artificial light at night) and Control (natural photoperiod) samples on experimental Day 49. Individual data points are plotted as circles. Boxes represent quartiles, and whiskers represent the range. Outliers are shown as darker-shaded circles. The figure was created using the R package ggplot2 v.3.4.2 (Wickham, 2016).

Finally, exploratory analyses showed a positive relationship between leaf count and total leaf area on Day 28 (Fig. 5A), with a strong correlation between these variables both for the ALAN group (r = 0.90, CI = [0.85, 0.94], p = 0.000) and for the Control group (r = 0.85, CI = [0.78, 0.91], p = 0.000). There was also a negative relationship between leaf count and the measure of dark pigmentation on the undersides of the leaves on Day 49 (Fig. 5B), with moderate correlation values for both the ALAN group (r = −0.49, CI = [−0.64, −0.30], p = 0.000) and for the Control group (r = −0.41, CI = [−0.57, −0.20], p = 0.0002).

Figure 5 Relationships between leaf counts of 80 ALAN (artificial light at night) and 80 Control (natural photoperiod only) duckweed samples and two other measured variables.

(A) Total leaf surface area on Day 29 of the experiment. (B) Levels of dark pigmentation on the underside of the leaves on Day 49 of the experiment. Figures were created using the R package ggplot2 v.3.4.2 (Wickham, 2016) and ggbeeswarm v.0.7.2 (Clarke, Sherrill-Mix & Dawson, 2023).

Discussion

Summary of main findings

Overall, our results indicate that duckweed plants exposed to ALAN tend to respond with enhanced growth rate, but potentially also some physiological stress. We also observed greater variability of growth in the ALAN group in terms of total leaf surface area and leaf counts. Finally, we found that the lower sides of duckweed leaves exposed to ALAN were often darker than in the control group, likely due to the accumulation of protective dark pigments, such as anthocyanins. The increase in dark pigmentation suggests that ALAN induces physiological stress-response mechanisms involved in protecting tissues from oxidative damage. Below, we discuss our findings in detail.

Increased duckweed growth under ALAN

Total leaf area per sample on Day 49 was larger in the ALAN group than in the Control group. This difference was also clear from day 28 for the duckweed growth and reproduction measured as the count of leaves per sample (Table 1; Fig. 3). This finding is in line with the work of Liu et al. (2018), who found higher dry mass production (an alternative measure of growth rate) in the same species (Landoltia punctata) under constant light. Increased dry mass was also found in Yin et al. (2015) using a different duckweed species, Lemna aequinoctialis, under continuous light exposure. However, a study by Gallego, Chien & Angeles Jr (2022) concluded that photoperiod had no effect on the relative growth rate of Landoltia punctata exposed to three different light sources. Yet, a possible factor explaining this result is that their experiment was only run over 16 days, with only three replicates per treatment combination, and the effects could be too small to be reliably detected.

Increased variability in duckweed growth under ALAN

The plant samples in the ALAN group were more variable for all growth measures (Figs. 3 and 4). For example, for total leaf counts on Day 49, there are four outliers in the ALAN group, the largest of which had 34 leaves, while the Control group’s largest value was 21 leaves. This indicates that ALAN treatment caused some of the plants to respond more strongly than other plants within the same treatment group, which could be due to genetic differences and/or phenotypic plasticity. It is known that many organisms become more variable under stressful conditions (Ghalambor et al., 2007). Further, we observed higher and more variable leaf death rates in the ALAN group compared to the control group (Table 1; Figs. 4A and 4B), which could also be attributed to increased stress. Stress-induced expression of variability in phenotypic traits can potentially influence adaptive changes leading to population-level adaptation to stressful conditions (Hoffmann & Hercus, 2000).

Increased pigmentation under ALAN

Plants in the ALAN group had more pigmentation on the underside of the leaves (Fig. 4F; Table 1). The dark pigment is most likely anthocyanin, which is known to protect tissues from oxidative damage (Agati et al., 2021). Anthocyanin has been found in duckweed fronds exposed to heavy metal, and organic pollutants (Jayasri & Suthindhiran, 2017; Sharma & Kaur, 2020). High light intensity can also trigger anthocyanin biosynthesis, helping plants cope with oxidative stress (Tattini et al., 2005). Our results indicate that artificial light exposure at night can also induce this type of stress response, despite the relatively low light intensity used. At the same time, we observed that samples with the highest proportion of dark-pigmented areas were growing slower than samples with less pigmented leaves in both ALAN and Control treatments (Fig. 5B). Thus, it is possible that excessive production of dark pigment may slow down the growth of duckweed plants.

Limitations

Our experiment had five limitations. First, the light intensity of our ALAN treatment could be considered relatively weak when compared to lights used in cities. For example, streetlights have a maximum of 4800 lux (Bennie et al., 2016), and the light we used had a maximum of 200 lux. However, the illumination quickly gets weaker further from the light source, so our experimental design could have been realistic for locations further away from strong light sources or plants growing in dimly lit areas at night. Second, we also did not have an automated system to turn on and off the lights at exactly the same time every day, which might have added some daily variation in exposures. Third, daily changes in weather made the system more vulnerable to uncontrolled events (variations in daylight intensity and temperature) that could have affected the results. However, these limitations also make our experiment closer to conditions in real urban environments. Fourth, we had only two holding boxes and a single ALAN light source in our experiment, which ideally should be replicated using multiple boxes and light sources. Finally, the software we used for quantifying the pigmentation levels of the underside of the leaf only measured the proportion of darker areas rather than the full-colour spectrum. However, our approach to measuring relative levels of dark-pigmented leaf areas was robust to differences in light conditions given the light exposure when taking the images of plants was not even, which could have influenced the results of colour spectrum analyses.

Future research

Future work can be done in at least three areas, building on our results. First, investigating responses of the spotted duckweed Landolita punctata to different light wavelengths, illumination levels, and exposure durations of ALAN. Second, by carrying out equivalent experiments across different strains and duckweed species. The dark pigmentation in the leaves of duckweed is a species-specific trait, with species differing in their ability to produce dark pigments (Tippery & Les, 2020). Thus, it would be important to test the stress responses of other species of duckweed exposed to ALAN in terms of the production of pigments and other mechanisms related to antioxidant defences. future research could also quantify responses at genomic and transcriptomic levels and look at the potential carry-over effects in progeny fronds. Finally, results of our study could be used in a meta-analysis alongside other works on this topic to create a bigger picture of the effects of ALAN on plant growth and physiology.

Conclusions

We found that Landolita punctata responded to the ALAN exposure group by increased growth, but also higher phenotypic variability and increased production of dark pigmentation in the leaves, indicating physiological stress. It is probably the first time such a combination of effects was observed in plants’ response to ALAN. The results from this work could inform and inspire future experiments on duckweed species, contributing to our understanding of the effects of light pollution in urban areas on plants.

We are grateful to Shinichi Nakagawa for providing statistical advice on implementing and interpreting Generalised Linear Mixed Models and for commenting on the draft manuscript. We also thank Kohaku Nakagawa-Lagisz for providing assistance with the manual handling of the samples during the experiment.

Additional Information and Declarations

Competing Interests

Author Contributions

Data Availability

The authors declare there are no competing interests.

Totoro Nakagawa-Lagisz conceived and designed the experiments, performed the experiments, analyzed the data, prepared figures and/or tables, authored or reviewed drafts of the article, and approved the final draft.

Malgorzata Lagisz conceived and designed the experiments, performed the experiments, analyzed the data, prepared figures and/or tables, authored or reviewed drafts of the article, and approved the final draft.

The following information was supplied regarding data availability:

The data and code are available at Zenodo: Totoro Nakagawa-Lagisz, & Malgorzata Lagisz. (2023). Data and code for ”Shining a Light on Duckweed: Exploring the Effects of Artificial Light at Night (ALAN) on Growth and Pigmentation” (v.1.0.0) [Data set]. Zenodo. https://doi.org/10.5281/zenodo.8017670

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
