# Peer review of "Shining a light on duckweed: exploring the effects of artificial light at night (ALAN) on growth and pigmentation"

_PeerJ, doi:10.7717/peerj.16371_

## Round 0.1 · original submission · Major Revisions

I completely agree with the reviewers that this manuscript lacks major experimental details and has critical experimental design issues. Based on these major drawbacks, and also aware of the fact that this work is mostly carried out by junior high school students, I kindly request the authors to review the comments of the reviewers and resubmit this manuscript after correcting/clarifying all the comments.

·

Basic reporting

The English expression of the manuscript is relatively clear, but the accuracy of the terms should be further improved. Related literature of the effects of ALAN on plant growth should be added, and the structure of the paper also needs to be adjusted. For example, the experimental light environment parameters should be present in Materials & Methods rather than Results. Furthermore, the presence of three corresponding authors in the author column is confusing.

Experimental design

Inappropriate nighttime ALAN can affect plant germination, flowering, tissue repair, leaf retention, bud breakage, and increase disease susceptibility. In this paper, the impact of nighttime ALAN on the number of wild derived dotted duck leaves and the effect of the leaf pigment are investigated. The results have certain reference value for evaluating the impact of nighttime pond and shoreline landscape lighting on the growth of Aquatic plant.However, there are still the following problems in the manuscript, which is recommended to revise:
Line125-128:
The description of LED artificial light source is incomplete, which should include the following information: the Correlated Colour Temperature of the light source (CCT 3000K), the General Color Rendering Index Ra, and the Special Color Rendering Index R9. Since plants are very sensitive to some wavelengths in the artificial light spectrum, the light spectrum distribution diagram and Peak Wavelength of the light source should also be given. Product and mall information should not appear in the description of light source parameters.
In addition, different light distributions can also affect the absorption, reflection of light by plants. It is recommended to add Light Distribution Curve and lighting design sketches to supplement the details.
Line128-129:
1、The parameters of day light near the window were not described, and the experimental season, illumination range, color temperature variation range, and spectral distribution of day light during the experimental period were not explained. Although in ‘Resluts, Lights on plants’ (line 235), manuscript showed that the average illuminance of daylight on the plants was 493lx, due to significant differences of daylight illuminance from morning to dusk, as well as significant differences of illuminance near windows in different directions, the average illuminance couldn't accurately describe the real state of daylight illumination. Authors did not provide any description of the season of the experiment, the orientation of the window, the transparency of the glass, the curtains, the daylight distribution in the area where the plants were located during the experiment, and the variation of illumination over time. Please provide the above information.
2、The daylight intensity and direct sunlight/reflection ratio vary at different positions on the window edge. Therefore, randomly changing the position of samples can result in parameters such as the number and wavelength of sunlight received by each sample unknowable. The intensity of daylight is hundreds or even thousands of times that of artificial light at night. Therefore, without a description of daylight, it is difficult to prove that changes in leaves are caused by artificial light at night. Only when the daylight environment of the experimental group and the control group during the day is consistent can it be proven that the changes in leaves are caused by artificial light at night. .
Line 136:
The main research content of this article is the impact of light on plant growth. Therefore, professional lighting measurement instruments such as Illuminometers and Colour Luminance Meters should be used. The manuscript did not introduce the type of lighting measurement equipment used in this experiment and the measurement methods for light environment.
Line 204:
“Lights on plants”, this part is too simple.

Validity of the findings

The paper lacks enough innovation, but it also provides a relatively convenient and inexpensive experimental method and measurement method for the study of the impact of nighttime landscape lighting on Aquatic plant. Therefore, this paper still has some value.

Additional comments

Comments can be found in Part 2. Experimental design.

Reviewer 2 ·

Basic reporting

The manuscript “Shining a light on duckweed: exploring the effects of Artificial Light at Night (ALAN) on growth and pigmentation“ by Totoro Nakagawa-Lagisz. and Malgorzata Lagisz describes an interesting study, aimed to understand the possible effects of ALAN on growth and stress tolerance of a duckweed species.
The authors three major findings are as follows; first, they found Landolita punctata responded to the ALAN exposure group by increased growth. They also found higher phenotypic variability - greater variability of growth in the ALAN group in terms of total leaf surface area and leaf counts. Finally, they found increased production of dark pigmentation in the leaves, an accumulation of anthocyanins, in the ALAN group. They interpreted these findings as indications of physiological stress; both positive effect (on growth) and negative effect (as a stressor) of ALAN.
This manuscript adds a nice piece of evidence for our knowledge of one of the major human-driven perturbations, ALAN, especially in urban ecosystems. The manuscript is well-written and the data is important. However, there are some difficulties in the experimental design that may jeopardize acceptance of this paper.
First, the experimental design exhibited limitations by not accounting for external variables, such as the precise positioning of the air conditioner and the potential influence of natural light from the window, which may have introduced confounding factors in the study's results. To enhance the rigor of the experimental design, it is imperative to address potential confounding factors by incorporating measures such as utilizing light sensors to automate light control, implementing temperature regulation mechanisms, and employing regular pH monitoring, thereby ensuring a more controlled and reliable research environment. The light source used in this research exhibit a maximum intensity of 200 lux, which deviates from the average luminance of streetlights. It is noted that luminous intensity diminishes significantly as the distance from the light source increases. To address this issue, a straightforward resolution would involve in-situ measurement of light intensity beneath streetlights and subsequently replicating, to the best extent possible, the same luminous intensity within their experimental setting.
Second, and probably the most difficult limitation of this study, the sample size and replication are of major concern. The experiment was done in two boxes, each contains 80-81 caps with plants. Each treatment was applied on all the plants in the box. Although the authors claim for n=80 sample size, because all plants of one treatment were in the same box, all the 80 plants are pseudo-replications and n=1 for each treatment.
Finally, although the introduction contains the relevant basis citations, I would like to see more biological background as for the physiological effect of ALAN on pigment accumulation and growth rate.

Experimental design

see above

Validity of the findings

see above.

Additional comments

Minor comments:
L.64: The word “stunted” is not trivial for non-English speakers and can be misleading. Replace with “
L.65: Change “to synthesize…cycle” to “for photosynthesis.”
L. 72: give botanical scientific name of the family.
L.76: I am not familiar with the word “frond”. Use a botanical term.
L78-81: Elaborate on the difference (spectrum) between ALAN and daylight. Elaborate what is missing in the cited studies. What is that we don’t know yet?
L. 83 It is already clear from the introduction light can be a stress
L. 84 Change to stressful abiotic conditions
L. 88: What heavy metals and pollution have to do with ALAN effect?
L. 95 Add hypotheses
L.98-99 and throughout the manuscript: The MeRIT system is not common enough and is a confusing way of writing. Replace all name acronyms with “we” and detail separately the contribution of each author to the study.
L.110: “daughter” and “granddaughter” may refer to generations, which are result of reproduction and genetic recombination. These words are not valid for vegetative reproduction.
L.124: how ventilation was allowed?
L.126: What is the light spectrum if the LED? How similar to (or different from) natural sun light?
L.129: Each box (with 80-81 cups) is a replicate. Total replicates per treatment are n=1.
L. 145-149: not clear how blindness was achieved.
L181-191: Note that darkness of the leaf is categorical (binomial) trait. Not quantitative. This is based on estimation of “dark” and “light”, of which the border between them is not defined. What is the threshold to define a leaf “dark”?
Statistical analysis: Most of the tests are not testing the major question. To account for weekly measurements, use time-series analysis. In any case, because n=1, most tests are not valid.
Results: All the results do not take into account the dependence between cups within treatment. These are not independent replicates.
L.287: The word “enhanced” refer to a non-significant result, so it is not certain that ALAN enhanced growth rate.
L. 291: Why is variance important? How this answer the research question? What is the biological relevance of variance per-se?
L.297: Note that number of leaves ≠ biomass.
L.313: Can you rule out genetic variance? Or because all plants are clonal, they are likely one genotype and the differences are plastic. This should be discussed.

---

## Round 0.2 · accepted · Accept

In this revised manuscript, most of the major issues were addressed. Since this study was carried out mainly by school students, it's hard to repeat this experiment for proper replication in an individual box as suggested by one of the reviewers. Since all the experimental details are now mentioned in detail, I will recommend them for publication.

·

Basic reporting

no comment

Experimental design

no comment

Validity of the findings

no comment

Additional comments

no comment

Reviewer 2 ·

Basic reporting

Most major issues were addressed. However, the major flaw in the experimental design is un-solvable. That is, the non-replicated treatments of 80 plants in each single box. Unfortunately, this n=1 issue preventing inference of the effect of ALAN on this plant.

Experimental design

see above

Validity of the findings

see above

Additional comments

see above